# Estimating the Impact of Climate and Vegetation Changes on Runoff Risk across the Hawaiian Landscape

Lucas Berio Fortini [1,*] , Lauren R. Kaiser [2] , Kim S. Perkins [3], Lulin Xue [4] and Yaping Wang [5]

1   U.S. Geological Survey (USGS), Honolulu, HI 96818, USA
2   Hawai'i Cooperative Studies Unit (HCSU), Honolulu, HI 96822, USA
3   U.S. Geological Survey (USGS), Menlo Park, CA 94025, USA
4   National Center for Atmospheric Research, Boulder, CO 80307, USA
5   SAIC at NOAA/NWS/NCEP/EMC, College Park, MD 20740, USA
*   Correspondence: lfortini@usgs.gov; Tel.: +1-808-230-3669

**Abstract:** In Hawai'i, ecosystem conservation practitioners are increasingly considering the potential ecohydrological benefits from applied conservation action to mitigate the degrading impacts of runoff on native and restored ecosystems. One determinant of runoff is excess rainfall events where rainfall rates exceed the infiltration capacity of soils. To help understand runoff risks, we calculated the probability of excess rainfall events across the Hawaiian landscape by comparing the probability distributions of projected rainfall frequency and land-cover-specific infiltration capacity. We characterized soil infiltration capacity based on different land cover types (bare soil, grasses, and woody vegetation) and compared them to the frequency of large rainfall events under current and future climate scenarios. We then mapped the potential risk of excess rainfall across the main Hawaiian Islands. Our results show that land cover type has a very large effect on runoff risk as excess rainfall conditions are 234% more likely in bare soil and 75% more likely in grasslands compared to woody forests and shrublands. In contrast, projected shifts in rainfall intensity by end-of-century show little impact on these probabilities and thus, the risk of runoff. This indicates that the probability of excess rainfall is primarily driven by differences in land cover and not by current or potential shifts in rainfall patterns across the Hawaiian landscape. The ability to estimate the risk of potentially ecologically and economically costly runoff based on changes of land cover is useful for managers focused on invasive species control and restoration planning, especially for native and endemic ecosystems unique to the State of Hawai'i.

**Keywords:** infiltration; excess rainfall; rainfall intensity; runoff; ecohydrology



## 1. Introduction

Native habitat loss, invasive species, and other threats have not only had large impacts on biodiversity of island systems but have likely greatly altered their ecohydrology. Reduced infiltration, the soil's ability to allow water through it, may lead to serious consequences including terrestrial habitat damage by erosion, aquatic habitat damage by sedimentation, and downstream damage by flooding due to storm flows from overland flow or runoff [1]. The relation between peak rainfall and runoff has been widely studied given its implications for local and downstream ecosystems and users [2–5]. Precipitation that is not transpired or evaporated back to the atmosphere can either run off, or infiltrate into the ground providing water for plants, base flow to streams, and recharge to aquifers. One of the primary mechanisms for runoff generation is Hortonian overflow, which occurs when intense storm events produce rainfall rates higher than the infiltration capacity of soils [6]. When this occurs, the excess rainfall may cause overflow water to run off over the surface. Hortonian overland flow typically occurs on steep slopes, in areas with impermeable soils, and when soils are already saturated and water tables are elevated. Hortonian runoff-driven flooding has been associated with large economic damage [7,8],

and in Hawai'i specifically runoff from excess rainfall has led to extensive erosion on Kaho'olawe [9].

The variability of soil infiltration, generally characterized as saturated hydraulic conductivity (Kfs), is considered a major determinant of runoff potential [10]. Land cover strongly affects the infiltration capacity of soils and has been shown to substantially affect runoff and hydrologic function [11,12]. Vegetation protects the soil from raindrops' kinetic energy, thus lowering its erosive capacity. It also enhances soil aggregation, or the arrangement of primary soil particles (sand, silt, clay), which impacts the structure and function of the soil [13]. However, questions related to soil infiltration have rarely been explored in tropical watersheds and ecosystems [14]. Previous research [15] showed that runoff generation in degraded Andean ecosystems is mainly controlled by the surface vegetation cover and land management. In those field-based experiments, degraded and abandoned land generated surface runoff within a few minutes after the start of the rainfall event and Hortonian overflow was found to be the dominant runoff generation mechanism. In West Africa, Hortonian-driven runoff has been shown to occur during intense storms and is a dominant process through which rainfall is redistributed over the landscape [14].

Various efforts to systematically evaluate surface hydrology in relation to vegetation type have been conducted at specific sites or for selective land cover type(s) or soil types in Hawai'i. In fact, the largest field effort to characterize soil water infiltration rates across forests in Hawai'i was recently completed [16]. That work explored the differences in key vegetation and soil characteristics that control runoff and groundwater recharge across managed, disturbed, and relatively intact native mesic and wet forest communities that provide important habitats to many native plant and animal species that are endemic to Hawai'i. In Hawai'i, mesic and wet forests are generally found in middle to upper elevations where mean annual rainfall is typically higher than 1200 mm and 2500 mm annually (mesic and wet forests, respectively; [17,18]). Kennedy, Mair, and Perkins [19] have conducted similar infiltration measurements across native and non-native forests in Maui. Both study designs complemented recent work by Perkins, Stock, and Nimmo [20], where they performed a survey and analysis of all past infiltration studies on other land cover types in Hawai'i. Together, there are now infiltration estimates across dry, mesic, and wet forests and shrublands, grasslands, and bare soil areas across the state.

These recent studies indicate that infiltration may vary remarkably across the landscape [16]. Beyond the wide range of infiltration capacity apparent in Hawaiian soils, rainfall gradients across the islands are some of the widest in the world. Mean annual rainfall ranges from as little as 204 mm (8 inches) to more than 10,200 mm (400 inches) across the landscape, and in many places mean annual rainfall can change by over 635 mm (25 inches) in the span of 1.5 km (1 mile) [21]. Due to this large variation in rainfall, a characterization of Hawaiian soil infiltration capacity without the context of rainfall values across such a variable landscape may be of limited use in assessing the probability of excess rainfall and consequently runoff risk. This link between rainfall and runoff based on vegetation-mediated infiltration has raised questions about the effects of climatic shifts on runoff as the distribution of vegetation and patterns of rainfall are subject to change [22,23]. However, because runoff events are often short-lived, related research requires the modeling of potential shifts in fine scale temporal frequency of rainfall events to then assess potential changes in runoff.

Despite the recent comprehensive characterization of infiltration across the Hawaiian landscape, it is still unclear how runoff risk across the state may be driven by vegetation differences and the wide landscape variability in rainfall [21]. With this in mind, we aimed to characterize the distribution of excess rainfall among broad vegetation communities and evaluate potential changes and trends in the future. We combined vegetation-dependent infiltration data with high-resolution gridded rainfall frequency estimates to provide fine-scale estimates of excess rainfall probability across the Hawaiian landscape as a measure of runoff risk. We conducted a frequency analysis of rainfall events in the context of vegetation-specific soil infiltration capacity to calculate runoff risk based on the probability

of excess rainfall events. We then developed maps illustrating the probability of excess rainfall events across the Hawaiian landscape under current and future climate scenarios. These results can be used to estimate shifts in runoff risk due to changes in climate and land cover or land use.

## 2. Materials and Methods

We used a landscape-scale approach to estimate the probability of excess rainfall as an indicator of runoff risk based on land cover type under current and future rainfall regimes. This was done by combining regionally downscaled estimates of hourly rainfall with infiltration measurements collected across three different cover classes (bare soils, grasslands, and woody vegetation).

### 2.1. Regionally Downscaled Climate Data

To develop estimates of hourly rainfall intensity for our research, we utilized fine-scale regional climate simulations for the main Hawaiian Islands recently developed by the National Center for Atmospheric Research (NCAR) [24]. These estimates of hourly rainfall intensity and frequency were derived using the Weather Research and Forecasting (WRF) model and are based on a 10-year simulation period for both a current (2002–2012) and future (2090–2099) representative concentration pathway (RCP) 8.5 scenario. This dataset had a major advantage of providing validated hourly rainfall values for the entire state, thus eliminating the need for us to statistically interpolate hourly or daily values from available monthly mean data. The historical simulation is based on the ERA-Interim global reanalysis data and observed sea surface temperature from October 2002 to September 2012 [25]. We used this hourly modeled dataset to characterize current rainfall intensity distributions as no spatial observation-based datasets were available at the time of our study. Nevertheless, the current rainfall simulations closely matched rainfall observations based on comparisons with weather station data [24]. The future projection uses the Pseudo Global Warming (PGW) method to implement change to historical conditions based on climate change signals from global climate model averages under RCP 8.5 emissions. This RCP is a greenhouse gas concentration trajectory that is intended to represent a hypothetical future scenario where greenhouse gases continue to increase without significant efforts to mitigate emissions based on plausible socioeconomic and technological trends [26,27]. For both current and future scenarios, we characterized the distribution of rainfall intensity by binning the hourly data on a logarithmic scale based on recurrence events by magnitude in millimeters per hour (mm/hr).

### 2.2. Regional Infiltration and Land Cover Data

The recent study by Berio Fortini et al. [16] characterized infiltration within mesic and wet forests and its uncertainties to an unprecedented detail, with more than 600 measurements spread across the Islands of Hawai'I and Kaua'i. This work complemented previous work in dry and non-forested environments [20,28,29] and additional forest infiltration measurements by the U.S. Geological Survey (USGS) Pacific Island Water Science Center [19]. We combined these infiltration measurements (saturated hydraulic conductivity, Kfs) and classified them based on three general land cover types: bare soil, grasses, and woody vegetation (shrubs and forests combined) to create infiltration rate estimates. Using these new estimates, we calculated when rainfall events are expected to exceed local cover-specific infiltration rates to determine the probability of excess rainfall.

We used the most up-to-date land cover map for the state [30] to classify different land cover types used for this study. The hierarchical classification system of this 30-m resolution land cover map allowed us to generalize various cover types into the three major land cover units of bare soil, grasses, and woody vegetation. This simplification of land cover types was made not only to match the classification of the infiltration dataset, but also to reflect the major land cover classifications and types of cover that are most prevalent where vegetation change occurs. All land cover data and maps were scaled to a 90 m

resolution. Areas classified as young lava flows (<10,000 years old), in which minimal soil development has occurred, and heavily disturbed or developed areas were excluded from this analysis [30].

### 2.3. Calculating Excess Rainfall Probabilities

Rainfall intensity probability distributions for the entire state and measurements of infiltration capacity, or Kfs, across the landscape were both measured in millimeters per hour (mm/hr), allowing for direct comparisons (Figure 1). We used the hourly rainfall intensities and compared these rainfall rates with the infiltration rates by land cover type. Considering both variability in vegetation cover and rainfall probability distributions across the landscape, we could use these two datasets to calculate the probability that rainfall events may exceed the infiltration capacity at each pixel across the landscape. The probability of excess rainfall was calculated based on the comparison of the probability distributions for rainfall and the vegetation cover specific infiltration capacity (Kfs). More specifically, for a particular pixel in the landscape, the probability of excess rainfall occurring for a specific rainfall intensity (e.g., 50 mm/h) is the probability of occurrence of such rainfall event multiplied by the probability that the infiltration rate is smaller than the given rainfall intensity (50 mm/h), based on the probability distribution of Kfs values for the vegetation class in that specific location. Furthermore, by calculating this probability over all rainfall intensity intervals for a given location, we can estimate the total probability of excess rainfall events for each point across the landscape. Using the future projections for rainfall intensities, we calculated equivalent excess rainfall probabilities for the future (2090–2099) RCP 8.5 scenario. All data processing and analyses were performed in R [31].

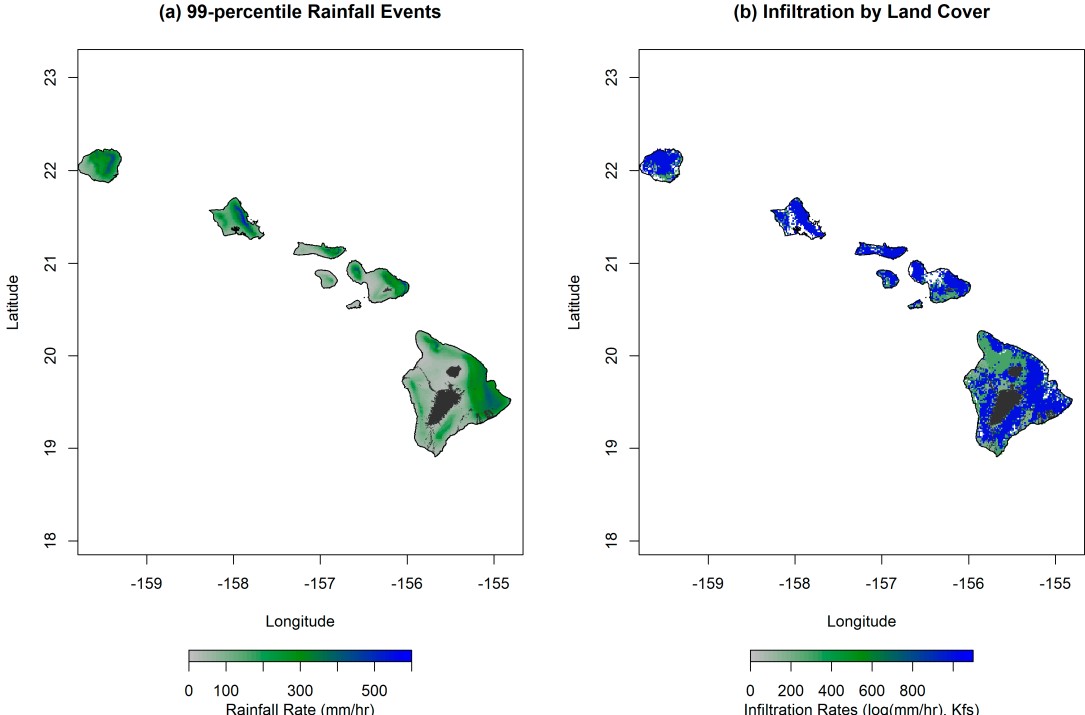

**Figure 1.** (**a**) Rainfall intensity (mm/hr) of 99th percentile events and (**b**) soil infiltration rates (Kfs) for Hawai'i. Areas in black indicate areas of young volcanic substrate that have been excluded from this analysis.

### 3. Results

Infiltration capacity of soils (Kfs) varied markedly between the three classes considered (176 ± 270 mm/h, 290 ± 97.9 mm/h, 1011 ± 147 mm/h; for bare soil, grassland, and woody cover, respectively), resulting in highly significant statistical differences determined by an

analysis of variance (ANOVA, F(2, 1000) = 17.36, *p*-value = 3.87e−08). A post-hoc analysis using Tukey's test revealed significant differences between woody cover and bare soil (*p*-value < 0.0001), and woody cover and grasslands (*p*-value < 0.0001), but no significant differences between bare soil and grasslands.

We mapped the probability of excess rainfall across the Hawaiian Islands (Figure 2). From our mapped results, we were able to identify differing excess rainfall probabilities associated with different land cover types. For example, areas with bare soils where the landscape has either been degraded or converted or new soils are being developed from previous lava flows had a probability of excess rainfall above 0.4. Conversely, areas associated with woody vegetation cover had a lower probability of excess rainfall (below 0.20). Differences in the probabilities of excess rainfall events across the state were primarily driven by the different types of land cover (Table 1). Excess rainfall conditions are 234% and 75% more likely in bare soil and grasslands, respectively, compared to forests and shrublands. The cumulative probability of infiltration capacity (in Kfs) by land cover type (Figure 3) further highlights the differences in infiltration between bare soil, grasslands, and woody vegetation. This can also be seen in detail when comparing the probability of excess rainfall at specific rainfall intensities (Figure 4) to individual land-cover-based infiltration rates. Finally, we projected future change in excess rainfall probabilities between current and the end-of-century RCP 8.5 scenario (Figure 5). Overall, climate scenario-based differences in excess rainfall were very small, ranging from a minimum 2.4% decrease to a 3.6% increase in the relative probabilities of excess rainfall events across the landscape.

**Table 1.** The mean probability of excess rainfall by land cover type across the State of Hawai'i at 90 m resolution.

| Land Cover Type | Mean Probability of Excess Rainfall | Standard Deviation | Number of Pixels |
|---|---|---|---|
| Bare Soil | 0.453 | 0.004 | 947 |
| Grassland | 0.236 | 0.002 | 1321 |
| Woody Vegetation | 0.135 | 0.001 | 4610 |

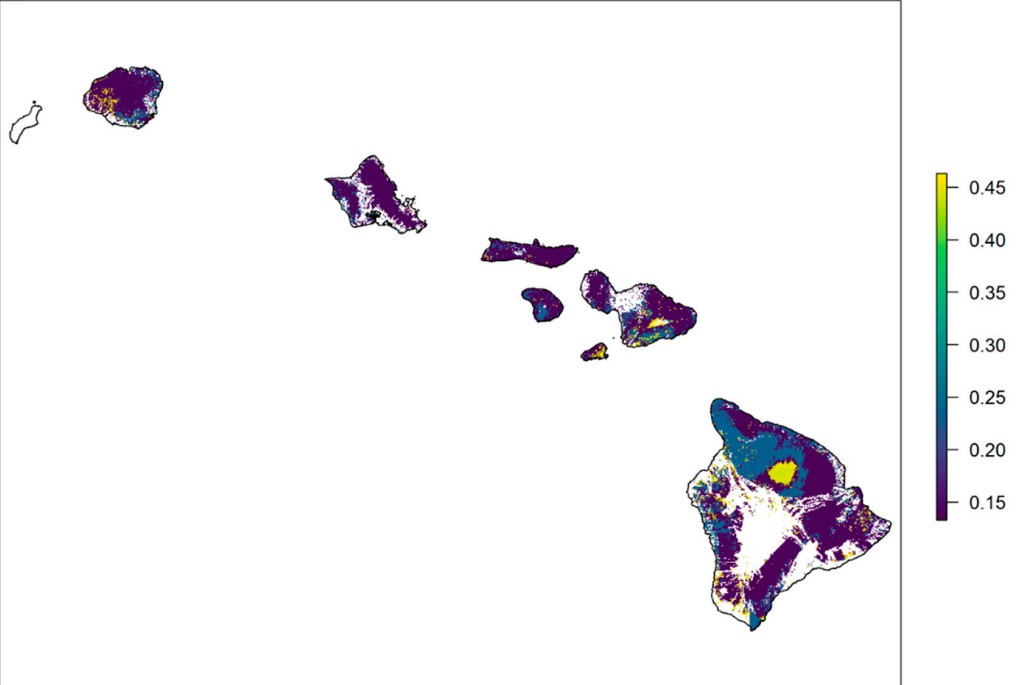

**Figure 2.** Probability of excess rainfall map for the Hawaiian Islands. Areas in white indicate areas of young volcanic substrate (<10,000 years old) or heavily disturbed/developed areas that were excluded from this analysis.

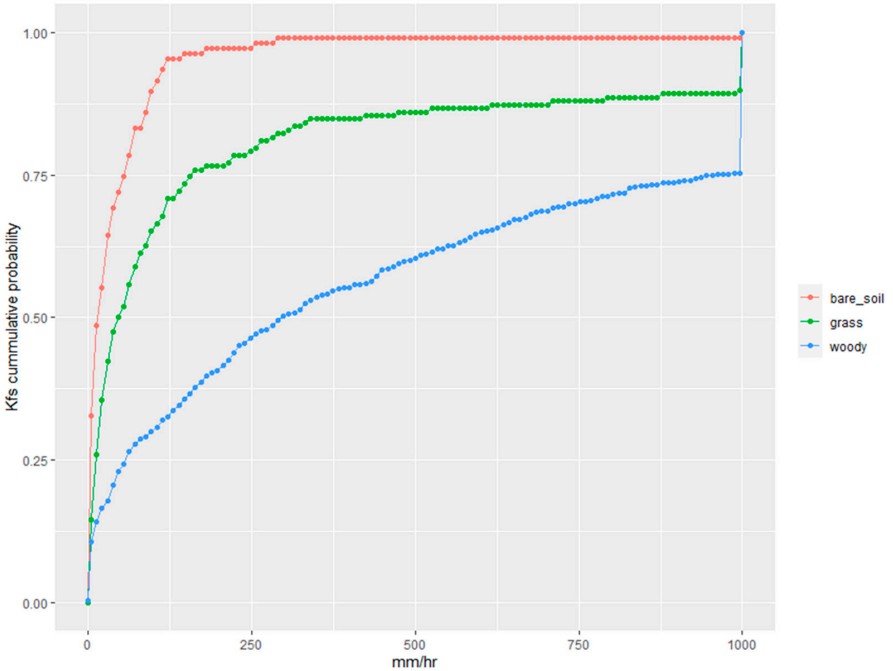

**Figure 3.** The cumulative probability of infiltration capacity (Kfs) of bare soil, grasses, and woody vegetation.

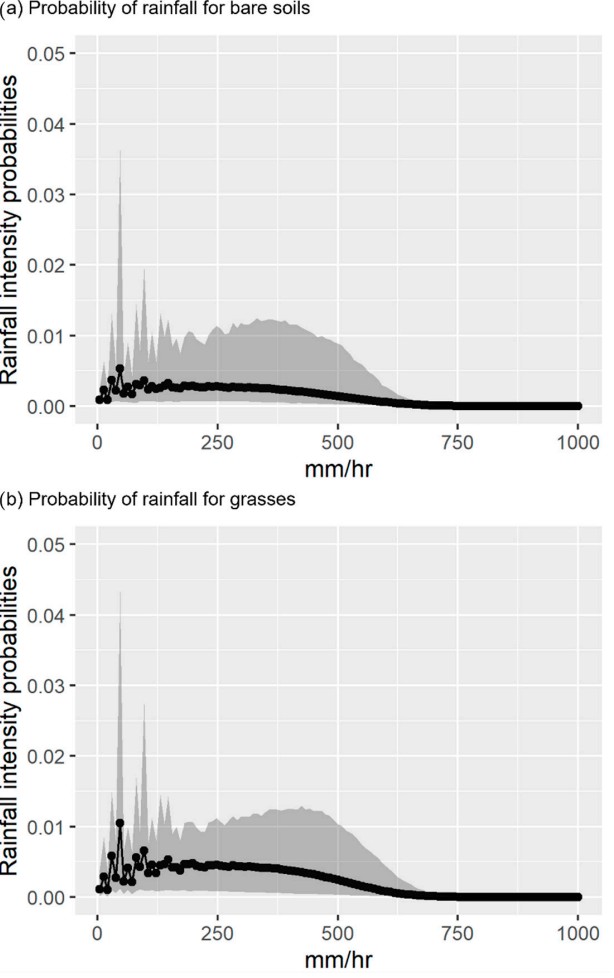

**Figure 4.** *Cont.*

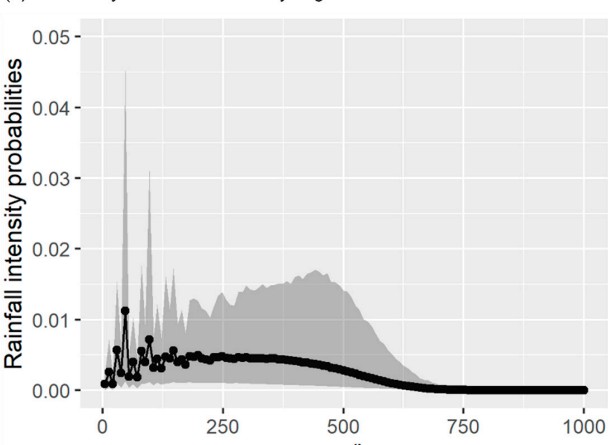

**Figure 4.** The probability of rainfall events and intensities of (**a**) bare soils, (**b**) grasses, and (**c**) woody vegetation. Black line indicates mean values, and the gray area indicates the range of values.

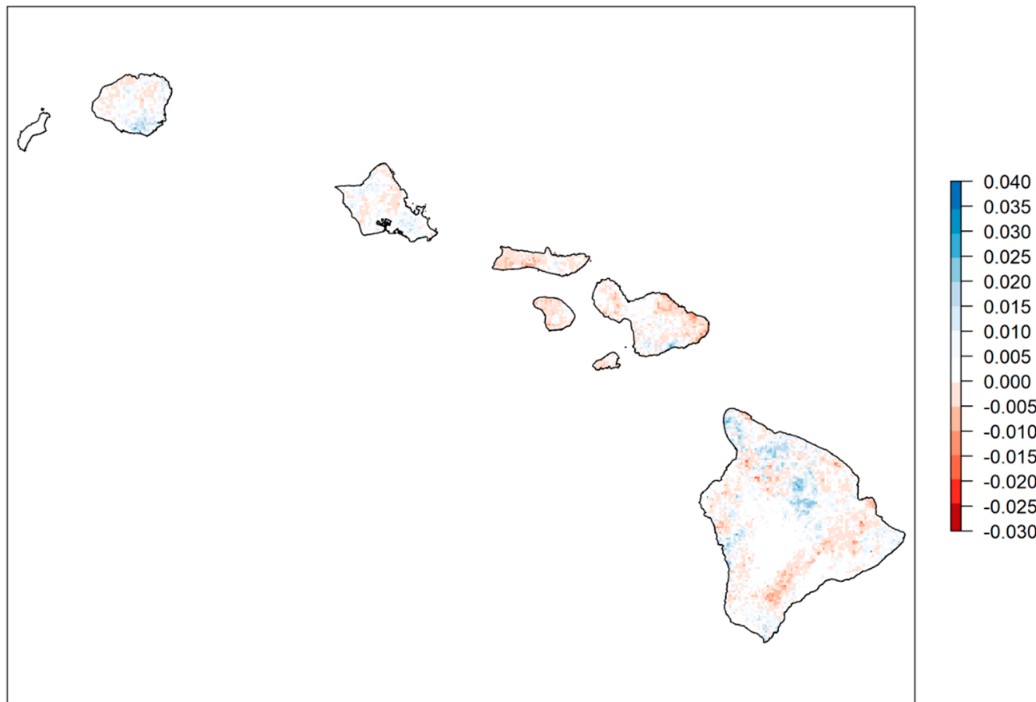

**Figure 5.** Proportional changes in excess rainfall probabilities between current and end-of-century RCP 8.5 scenario.

As a partial validation of our results, we compared our excess rainfall probabilities with runoff to rainfall ratios derived from a recent study on Maui [32]. To do that, we calculated mean excess rainfall probabilities across the same watersheds in which runoff to rainfall ratios were calculated, resulting in a moderate correlation between the two metrics ($r^2 = 0.48$), despite a mismatch in spatial coverage and baseline periods between studies (2002–2012 for our study versus 1978–2007 for the runoff study [32]).

## 4. Discussion

Although we did not measure runoff directly, when rainfall rates exceed the infiltration capacity of different vegetation cover categories, the excess rainfall may flow over land as runoff. Consequently, combined with past research, our results show that vegetation cover has a very large effect on infiltration rates, excess rainfall probabilities, and thus, likely

runoff. The differences in probability of excess rainfall across the three broad vegetation cover categories considered (bare soil, grasses, and woody vegetation) was large, where areas with woody vegetation cover had higher infiltration rates and lower excess rainfall probabilities, followed by areas classified as grasses land cover and then bare soil.

Unexpectedly, our results indicate that the point at which excess rainfall exceeds infiltration that can result in runoff is driven primarily by differences in land-cover-driven infiltration capacity and not the differences in rainfall intensities seen across the Hawaiian Islands. The characteristics of the soil and vegetation play an important role in runoff generation [33,34]. For example, soil that is more compacted or has less organic matter has less infiltration capacity and hence can produce more excess rainfall and potential runoff. Similarly, areas with less vegetation cover can produce more runoff because there are fewer plants to absorb and transpire water. While rainfall patterns are a factor in determining individual runoff events [35–37], we show vegetation cover plays a critical role in reducing runoff, for instance by intercepting rainwater and promoting infiltration [38–41]. However, it is also important to note that vegetation cover itself is dependent on rainfall patterns [42] where, for example, grassland areas typically occur in more arid areas than forested areas.

As large changes in land cover across the landscape continue [43,44], our research offers the ability to broadly estimate potential consequences of important factors such as infiltration capacity and excess rainfall probabilities that determine runoff across the landscape. Most notably from our results, the projected shifts in rainfall intensity by end-of-century show little effect on these probabilities and associated runoff risks. However, we were only able to consider a single future climate scenario, which is insufficient to describe the uncertainty characteristic of future rainfall projections for Hawai'i [45–47].

Our findings have potential implications for conservation and management practices in the Hawaiian Islands. Quantifying and understanding the differences in excess rainfall probabilities and related runoff risks can provide valuable information to managers responsible for maintaining watersheds, ecosystems, and downstream human needs. One of the key insights from our research is the importance of maintaining and restoring vegetation cover, especially forests and shrublands, to enhance infiltration capacity and reduce the probability of excess rainfall and associated runoff risks. A specific management implication of our work is to support the restoration efforts in areas with degraded or converted lands, which show a higher risk of runoff. By focusing on these areas, land managers can further reduce additional risk such as soil erosion, flooding, and water pollution caused by runoff [48]. A peculiarity of Hawaiian vegetation is that prior to the introduction of multiple invasive grasses there was no substantial native grassland cover across the landscape [49–51]. Significant effort has been spent in the state to limit the spread of these invasive grasslands to prevent increasing fire risk across the landscape and other detrimental ecological impacts. Our research further supports these management efforts by showing that the preservation of woody cover also includes reducing the risk of excess rainfall conditions and the additional associated runoff risks.

Lastly, and on a broader level, considering effects of vegetation cover and climate change on ecohydrology of the Hawaiian landscape essentially connects management efforts of one parcel to its surrounding stakeholders. This is important as recent research illustrates that climate-driven ecological shifts in Hawai'i will likely require effective management across boundaries [52]. Although our results provide a novel product available across the landscape, there are several caveats to be considered. First, our field measurements of infiltration only considered the saturated infiltration capacity of soils, meaning that we did not consider infiltration rates of different saturation levels. Second, although we were able to characterize the probability of excess rainfall, this was done solely based on local infiltration rates of different vegetation cover types, discounting the initial depression storage (where low-lying areas in uneven terrain can temporarily hold rainfall that would otherwise be runoff), which is difficult to estimate at a landscape level [53–55]. Even though we did not physically measure the depression storage capacity, preliminary topographic analyses showed that it was likely relatively minor across the steep Hawaiian landscape.

Our analysis also does not consider saturated excess rainfall based on antecedent rainfall or Dunne flow, which may be an equally important mechanism for determining runoff risks [56,57]. Finally, the source of infiltrating water is not limited to direct rainfall. Runoff generated upslope can percolate through permeable areas downslope, a process termed run-on infiltration [58]. Given the high variability of infiltration within each cover type we considered, this process likely plays a role in mitigating runoff risks across the Hawaiian landscape.

## 5. Conclusions

Our study provides valuable insights into the role of vegetation cover and its influence on infiltration capacity, excess rainfall probabilities, and runoff risks across the Hawaiian landscape. We found that woody vegetation cover exhibited higher infiltration rates and lower probabilities of excess rainfall, followed by grasses and then bare soil. These findings highlight the important role of vegetation cover in determining runoff risks, emphasizing the importance of preserving and restoring native vegetation, particularly forests and shrublands.

The potential impacts of climate change on rainfall patterns and runoff risks were also considered, revealing that the projected shifts in rainfall intensity by end-of-century have unexpectedly little effect on excess rainfall probabilities and associated runoff risks under the single climate scenario studied. This underlines how important it is for further research to consider multiple climate scenarios to better understand the uncertainty surrounding future rainfall projections for Hawai'i.

Our study offers useful information for land managers, conservationists, and city planners alike in the Hawaiian Islands to better understand the relation between vegetation cover, climate change, and ecohydrological processes. The ability to estimate the risk of runoff based on changes of land cover may be especially useful for managers focused on invasive species control and restoration planning, especially for native and endemic ecosystems unique to the State of Hawai'i. Further research exploring multiple climate scenarios, different saturation levels of infiltration, and other contributing factors to runoff risks, such as depression storage and Dunne flow, would be helpful in refining our understanding of runoff generation and informing future management strategies in the region.

**Author Contributions:** Conceptualization, L.B.F.; methodology and formal analysis, L.B.F. and L.R.K.; data curation, L.B.F., K.S.P., L.X. and Y.W.; writing—original draft preparation, L.B.F. and L.R.K.; writing—review and editing, K.S.P., L.X. and Y.W.; visualization, L.B.F. and L.R.K.; funding acquisition, L.B.F. All authors have read and agreed to the published version of the manuscript.

**Funding:** Funding and support for this project was provided by the U.S. Geological Survey (USGS) Pacific Island Climate Adaptation Science Center (PICASC), project award #COA.16DIR.USGS.LFO.01. This research was partially supported by the (USGS) Water Mission Area's Integrated Water Prediction Program. National Center for Atmospheric Research (NCAR) is a major facility sponsored by the National Science Foundation (NSF) under Cooperative Agreement 1852977. Any use of trade, firm, or product names is for descriptive purposes only and does not imply endorsement by the U.S. Government.

**Institutional Review Board Statement:** Not applicable.

**Data Availability Statement:** The datasets generated during and/or analysed during the current study are available at https://doi.org/10.5066/P9VOTDH3 [59].

**Conflicts of Interest:** The authors declare no conflict of interest. The authors have no relevant financial or non-financial interests to disclose. The funders had no role in the design of the study; in the collection, analyses, or interpretation of data; in the writing of the manuscript; or in the decision to publish the results.

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
