# Peer review of "Estimating the Impact of Climate and Vegetation Changes on Runoff Risk across the Hawaiian Landscape"

_conservation, doi:10.3390/conservation3020020_

Round 1

Reviewer 1 Report

Quantifying and understanding the differences in runoff can provide essential information to managers responsible for maintaining watersheds, ecosystems, and downstream anthropogenic needs. the results provide a novel product available across the landscape. However, the developped technic measure the saturated infiltration capacity only. Second,  the probability of excess rainfall is calculated without discounting the initial depression storage.

The developed method uses existing products. Therefore, the explanations of the method must be improved.

From the initial hypotheses, I invite the authors to compare the results obtained with the reality on the ground and to give the error parameters found. This will give the reader more information about the effectiveness of the developed product.

Reviewer 2 Report

It's meaningful and valuable.

there are two questions and suggestion:

(1)please give physical explanation to the result that  runoff risk  is not determined by rainfall pattens and intensities;

(2)in the line 161,how to calculate the probability...?

(3)in the figure 1,please give the name  of the abscissa and ordinate

Reviewer 3 Report

Thank you for providing the opportunity to review this manuscript which focuses determining runoff risk for varying cover types with climate and vegetation changes. Overall, I felt the manuscript was well written and the study design, well done. I appreciate the level of analysis, but I don’t see how managers can directly use the runoff probabilities, as stated in the Discussion section. Please provide a specific case or rationale.

Minor comments:

Line 56: Explain Hortonian runoff.

Line 118: Is the continued “high level emissions” what is known as “Business-as-unusual? If so, please clarify as continued high emissions could be viewed as an emissions scenario where high emissions occur beyond what the globe is now experiencing.

Line 22: what is mesic?

Lines 161 through 164: Why did you choose 0.4 and 0.2 as triggers?

Figure 3: Is the Y axis the probability of infiltration?

Figure 4: Why would rainfall intensity be different over bare soil, grass and woody debris for that matter? There is no canopy over bare soil and grass.

Line 198: Remove “do”

Line 198: When you say “excess rainfall” are you referring to runoff? How would the cover type impact the level of precipitation?
